# URLB: Unsupervised Reinforcement Learning Benchmark

**Michael Laskin**[*]
UC Berkeley
mlaskin@berkeley.edu

**Denis Yarats**[*]
NYU, FAIR
denisyarats@cs.nyu.edu

**Hao Liu**
UC Berkeley

**Kimin Lee**
UC Berkeley

**Albert Zhan**
UC Berkeley

**Kevin Lu**
UC Berkeley

**Catherine Cang**
UC Berkeley

**Lerrel Pinto**
NYU

**Pieter Abbeel**
UC Berkeley, Covariant

## Abstract

Deep Reinforcement Learning (RL) has emerged as a powerful paradigm to solve a range of complex yet specific control tasks. Yet training generalist agents that can quickly adapt to new tasks remains an outstanding challenge. Recent advances in unsupervised RL have shown that pre-training RL agents with self-supervised intrinsic rewards can result in efficient adaptation. However, these algorithms have been hard to compare and develop due to the lack of a unified benchmark. To this end, we introduce the Unsupervised Reinforcement Learning Benchmark (URLB). URLB consists of two phases: reward-free pre-training and downstream task adaptation with extrinsic rewards. Building on the DeepMind Control Suite, we provide twelve continuous control tasks from three domains for evaluation and open-source code for eight leading unsupervised RL methods. We find that the implemented baselines make progress but are not able to solve URLB and propose directions for future research. Code for the benchmark and implemented baselines can be accessed at https://github.com/rll-research/url_benchmark.

## 1 Introduction

Deep Reinforcement Learning (RL) has been at the source of a number of breakthroughs in autonomous control over the last five years. RL algorithms have been used to train agents to play Atari video games directly from pixels [44, 45], learn robotic locomotion [52–54] and manipulation [2] policies from raw sensory input, master the game of Go [58, 59], and play large-scale multiplayer video games [6, 65]. While these results were significant advances in autonomous decision making, a deeper look reveals a fundamental limitation. The above algorithms produced agents capable of only solving the single task they were trained to solve. As a result, current RL approaches produce brittle policies with poor generalization capabilities [16], which limits their applicability to many problems of interest [23]. It is therefore important to move beyond today's powerful but narrow RL systems toward generalist systems capable of quickly adapting to new downstream tasks.

In contrast, in the fields of Computer Vision (CV) and Natural Language Processing (NLP), large-scale unsupervised pre-training has enabled sample-efficient few-shot adaptation. In NLP, unsupervised sequential modeling has produced powerful few-shot learners [8, 17, 50]. In CV, unsupervised representation learning techniques such as contrastive learning have produced algorithms that are dramatically more label-efficient than their supervised counterparts [14, 31, 32, 25] and more capable of adapting to a host of downstream supervised tasks such as classification, segmentation, and object detection. While these advances in unsupervised learning have also benefited RL in terms of learning

---

[*]equal contribution, order determined by coin flip.

35th Conference on Neural Information Processing Systems (NeurIPS 2021) Track on Datasets and Benchmarks.

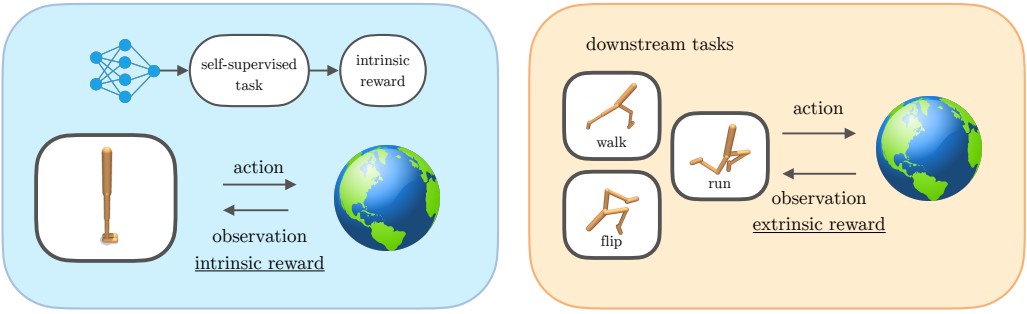

Part 1: Unsupervised Pre-training
with Intrinsic Rewards

Part 2: Supervised Finetuning
to Downstream Tasks

Figure 1: Unlike supervised RL which requires reward interaction at every step, unsupervised RL has two phases: (i) reward-free pre-training and (ii) fine-tuning to an extrinsic reward. During phase (i) an agent explores the environment through reward-free interaction with the environment. The quality of exploration depends on the intrinsic reward that the agent sets for itself. During phase (ii) the quality of pre-training is evaluated by its adaptation efficiency to a downstream task.

efficiently from images [37, 38, 55, 62, 69] as well as introducing new architectures for RL [13, 35], the resulting agents have remained narrow since they still optimize a single extrinsic reward as before.

Fully unsupervised training of RL algorithms requires not only learning self-supervised representations but also learning policies without access to extrinsic rewards. Recently, unsupervised RL algorithms have begun to show progress toward more generalist systems by training policies without extrinsic rewards. Exploration with self-supervised prediction has enabled agents to explore video games from pixels [48, 49], mutual information-based approaches have demonstrated self-supervised skill discovery and generalization to downstream tasks in continuous control domains [21, 30, 43, 57], and maximal entropy RL has yielded policies capable of diverse exploration [42, 56, 68]. However, comparing and developing new algorithms has been challenging due to a lack of a unified evaluation benchmark. Reward-free RL algorithms often use different optimization schemes, different tasks for evaluation, and have different evaluation procedures. Additionally, unlike more mature supervised RL algorithms [27, 33, 54], there does not exist a unified codebase for unsupervised RL that can be used to develop new methods quickly.

To make benchmarking and developing new unsupervised RL approaches easier, we introduce the Unsupervised Reinforcement Learning Benchmark (URLB). Built on top of the widely adopted DeepMind Control Suite [64], URLB provides a suite of domains of varying difficulty for unsupervised pre-training with diverse downstream evaluation tasks. URLB standardizes evaluation of unsupervised RL algorithms by defining fixed pre-training and fine-tuning procedures across all baselines. Perhaps most importantly, we open-source code for URLB environments as well as 8 leading baselines that represent the main approaches taken towards unsupervised pre-training in RL to date. Unlike prior code releases for unsupervised RL, URLB uses the same exact optimization algorithm for each baseline which enables transparent benchmarking and lowers the barrier to entry for developing new algorithms. We summarize the main contributions of this paper below:

1. We introduce URLB, a new benchmark for evaluating unsupervised RL algorithms, which consists of three domains and twelve continuous control tasks of varying difficulty to evaluate the adaptation efficiency of unsupervised RL algorithms.

2. We open-source a unified codebase for eight leading unsupervised RL algorithms. Each algorithm is trained with the same optimization backbone for fairness of comparison.

3. We find that while the implemented baselines make progress on the proposed benchmark, no existing unsupervised RL algorithm can solve URLB, and consequently identify promising research directions to progress unsupervised RL.

The benchmark environments, algorithmic baselines, and pre-training and evaluation scripts are available at `https://github.com/rll-research/url_benchmark`. We believe that URLB will make the development of unsupervised RL agents easier and more transparent by providing a unified

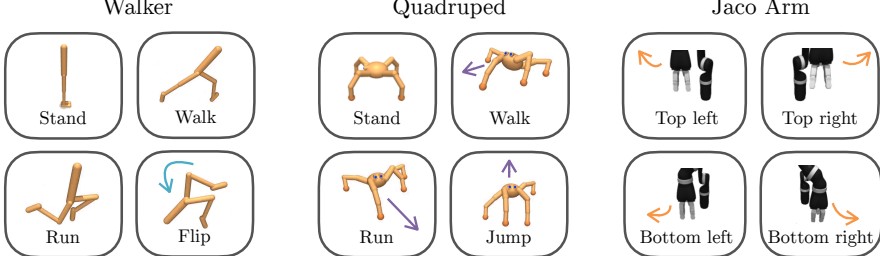

Figure 2: The three domains (walker, quadruped, jaco arm) and twelve downstream tasks considered in URLB. The environments include tasks of varying complexity and require an agent pre-trained on a given domain to adapt efficiently to the downstream tasks within that domain.

set of evaluation environments, systematic procedures for pre-training and evaluation, and algorithmic baselines that share the same optimization backbone.

## 2 Preliminaries and Notation

*Markov Decision Process:* We consider the typical Reinforcement Learning setting where an agent's interaction with the environment is modeled through a Markov Decision Process (MDP) [63]. In this work, we benchmark unsupervised RL algorithms in both fully observable MDPs where the agent learns from coordinate state as well as partially observable MDPs (POMDPs) where the agent learns from partially observable image observations. For simplicity we refer to both image and state-based observations as $\mathbf{o}$. At every timestep $t$, the agent sees an observation $\mathbf{o}_t$ and selects an action $\mathbf{a}_t$ based on its policy $\mathbf{a}_t \sim \pi_\theta(\cdot|\mathbf{o}_t)$. The agent then sees the next observation $\mathbf{o}_{t+1}$ and an extrinsic reward $r_t^{\text{ext}}$ provided by the environment (supervised RL) or an intrinsic reward $r_t^{\text{int}}$ defined through a self-supervised objective (unsupervised RL). In this work, we pre-train agents with intrinsic rewards $r_t^{\text{ext}}$ and fine-tune them to downstream tasks with extrinsic rewards $r_t^{\text{ext}}$. Some algorithms considered in this work condition the agent on a learned task vector which we denote as $\mathbf{w}$.

*Learning from pixels vs states:* We benchmark unsupervised RL where environment observations $\mathbf{o}_t$ can be either proprioceptive states or RGB images. When learning from pixels, rather than defining the self-supervised task directly as a function of image observations, it is usually more convenient to first embed the image and compute the intrinsic reward as a function of these lower dimensional features [10, 42, 43, 48]. We therefore define an embedding as $\mathbf{z}_t = f_\xi(\mathbf{o}_t)$ where $f_\xi(\mathbf{o}_t)$ is an encoder function. We employ different encoder $f_\xi$ architectures depending on whether the algorithm receives pixel or state-based input. For pixel-based inputs we use the convolutional encoder architecture from SAC-AE [66], while for state-based inputs we use the identity function by default unless the unsupervised RL algorithm explicitly specifies a different encoding. The intrinsic reward $r_t^{\text{int}}$ can be a function of any and all $(\mathbf{z}_t, \mathbf{a}_t, \mathbf{w}_t)$ depending on the algorithm. Finally, note that the encoder $f_\xi$ may or may not be shared with components of the base RL algorithm such as the actor and critic.

## 3 URLB: Evaluation and Environments

### 3.1 Standardized of Pre-training and Fine-tuning Procedures

One reason why unsupervised RL has been hard to benchmark to date is that there is no agreed upon procedure for training and evaluating unsupervised RL agents. To this end, we standardize pre-training, fine-tuning, and evaluation in URLB. We split pre-training and fine-tuning into two phases consisting of $N_{PT}$ and $N_{FT}$ environment steps respectively. During pre-training, we checkpoint agents at 100k, 500k, 1M, 2M steps in order to evaluate downstream performance as a function of pre-training steps. For adapting the pre-trained policy to downstream tasks, we evaluate in the data-efficient regime where $N_{FT}$ is 100k, since we are interested in agents that are quick to adapt.

## 3.2 Evaluation

We evaluate the performance of an unsupervised RL algorithm by measuring how quickly it adapts to a downstream task. For each fine-tuning task, we initialize the agent with the pre-trained network parameters, fine-tune the agent for 100k steps and measure its performance on the downstream task. This evaluation procedure is similar to how pre-trained networks in CV and NLP are fine-tuned to downstream tasks such as classification, object detection, and summarization. There exist other means of evaluating the quality of pre-trained RL agents such as measuring the diversity of data collected during exploration or zero-shot generalization of goal-conditioned agents. However, it is challenging to produce a general method to measure data diversity, and while zero-shot generalization with goal-conditioned agents can be powerful such a benchmark would be limited to goal-conditioned RL. For these reasons, data diversity and goal-conditioned zero-shot generalization are less common evaluation metrics. In an effort to provide a general benchmark, we focus on the fine-tuning efficiency of the agent after pre-training which allows us to evaluate a diverse set of baselines.

Unlike unsupervised methods in CV and NLP which focus solely on representation learning, unsupervised pre-training in RL requires both representation learning and behavior learning. For this reason, URLB benchmarks performance for both state-based and pixel-based agents. Benchmarking both state and pixel-based RL separately is important because it allows us to decouple unsupervised behavior learning from unsupervised representation learning. In state-based RL, the agent receives a near-optimal representation of the world through coordinate states. Evaluating state-based unsupervised RL agents allows us to isolate unsupervised behavior discovery without worrying about representation learning as confounding factor. Evaluating pixel-based unsupervised RL agents provides insight into how representations and behaviors can be learned jointly.

## 3.3 URLB Environments

We release a set of domains and downstream tasks for URLB that are based on the DeepMind Control Suite (DMC) [64]. The three reasons for building URLB on top of DMC are (i) DMC is already widely adopted and familiar to RL practitioners; (ii) DMC environments can be used with both state and pixel-based inputs; (iii) DMC features environments of varying difficulty which is useful for designing a benchmark that contains both challenging and feasible tasks. URLB evaluates performance on 12 continuous control tasks (3 domains with 4 downstream tasks per domain). From easiest to hardest, the URLB domains and tasks are:

**Walker** *(Stand, Walk, Flip, Run)*: A biped constrained to a 2D vertical plane. Walker is a challenging introduction domain for unsupervised RL because it requires the unsupervised agent to learn balancing and locomotion skills in order to fine-tune efficiently. **Quadruped** *(Stand, Walk, Jump, Run)*: A quadruped within a a 3D space. Like walker, quadruped requires the agent to learn to balance and move but is harder due to a high-dimensional state and action spaces and 3D environment. **Jaco Arm** *(Reach top left, Reach top right, Reach bottom left, Reach bottom right)*: Jaco Arm is a 6-DOF robotic arm with a three-finger gripper. This environment tests the unsupervised RL agent's ability to control the robot arm without locking and perform simple manipulation tasks. It was recently shown that this environment is particularly challenging for unsupervised RL [68].

# 4 URLB: Algorithmic Baselines for Unsupervised RL

In addition to introducing URLB, the other primary contribution of this work is open-sourcing a unified codebase for eight leading unsupervised RL algorithms. To date, unsupervised RL algorithms have been hard to compare due to confounding factors such as different evaluation procedures and optimization schemes. While URLB provides standardized pre-training, fine-tuning, and evaluation procedures, current algorithms are hard to compare since they rely on different optimization algorithms. For instance, Curiosity [48] utilizes PPO [54] while APT [42] uses SAC [27] for optimization. Moreover, even if two unsupervised RL methods use the same optimization algorithm, small differences in implementation can result in large performance differences that are independent of the pre-training algorithm. For this reason, it is important to provide a unified codebase with identical implementations of the optimization algorithm for each baseline. Providing such a unified codebase is one of the main contributions of this benchmark.

---

**Algorithm 1** Unsupervised RL: Unsupervised Pre-training and Supervised Fine-tuning

---
**Require:** Randomly initialized actor $\pi_\theta$, critic $Q_\phi$, and encoder $f_\xi$ networks, replay buffer $\mathcal{D}$.
**Require:** Intrinsic $r^{\text{int}}$ and extrinsic $r^{\text{ext}}$ reward functions, discount factor $\gamma$.
**Require:** Environment (env), $M$ downstream tasks $T_k, k \in [1, \ldots, M]$.
**Require:** pre-train $N_{\text{PT}}$ and fine-tune $N_{\text{FT}}$ steps.
1: **for** $t = 1..N_{\text{PT}}$ **do**                                   ▷ Part 1: Unsupervised Pre-training
2:      $\mathbf{a}_t \leftarrow \pi_\theta(f_\xi(\mathbf{o}_t)) + \epsilon$ and $\epsilon \sim \mathcal{N}(0, \sigma^2)$
3:      $\mathbf{o}_{t+1} \sim P(\cdot|\mathbf{o}_t, \mathbf{a}_t)$
4:      $\mathcal{D} \leftarrow \mathcal{D} \cup (\mathbf{o}_t, \mathbf{a}_t, \mathbf{o}_{t+1})$
5:      Update $\pi_\theta$, $Q_\phi$, and $f_\xi$ using minibatches from $\mathcal{D}$ and intrinsic reward $r^{\text{int}}$ according to Eqs. 1 and 2.
6: **end for**
7: Outputs pre-trained parameters $\theta_{\text{PT}}$, $\phi_{\text{PT}}$, and $\xi_{\text{PT}}$
8: **for** $T_k \in [T_1, \ldots, T_M]$ **do**                           ▷ Part 2: Supervised Fine-tuning
9:      initialize $\theta \leftarrow \theta_{\text{PT}}, \phi \leftarrow \phi_{\text{PT}}, \xi \leftarrow \xi_{\text{PT}}$, reset $\mathcal{D}$
10:      **for** $t = 1..N_{\text{FT}}$ **do**
11:          $\mathbf{a}_t \leftarrow \pi_\theta(f_\xi(\mathbf{o}_t)) + \epsilon$ and $\epsilon \sim \mathcal{N}(0, \sigma^2)$
12:          $\mathbf{o}_{t+1}, r_t^{\text{ext}} \sim P(\cdot|\mathbf{o}_t, \mathbf{a}_t)$
13:          $\mathcal{D} \leftarrow \mathcal{D} \cup (\mathbf{o}_t, \mathbf{a}_t, r_t^{\text{ext}}, \mathbf{o}_{t+1})$
14:          Update $\pi_\theta$, $Q_\phi$, and $f_\xi$ using minibatches from $\mathcal{D}$ according to Eqs. 1 and 2.
15:      **end for**
16:      Evaluate performance of RL agent on task $T_k$
17: **end for**

---

## 4.1 Backbone RL Algorithm

Since most of the above algorithms rely on off-policy optimization (and some cannot be optimized on-policy at all), we opt for a state-of-the-art off-policy optimization algorithm. While SAC [27] has been the de facto off-policy RL algorithm for many RL methods in the last few years, it is prone to suffering from policy entropy collapse. DrQ-v2 [67] recently showed that using DDPG [41] instead of SAC as a learning algorithm leads to a more robust performance on tasks from DMC. For this reason, we opt for DrQ-v2 [67] as our base optimization algorithm to learn from images, and DDPG, as implemented in DrQ-v2, to learn from states. DDPG is an actor-critic off-policy algorithm for continuous control tasks. The critic $Q_\phi$ minimizes the Bellman error

$$\mathcal{L}_Q(\phi, \mathcal{D}) = \mathbb{E}_{(\mathbf{o}_t, \mathbf{a}_t, r_t, \mathbf{o}_{t+1}) \sim \mathcal{D}} \left[ \left( Q_\phi(\mathbf{o}_t, \mathbf{a}_t) - r_t - \gamma Q_{\bar{\phi}}(\mathbf{o}_{t+1}, \pi_\theta(\mathbf{o}_{t+1})) \right)^2 \right], \qquad (1)$$

where $\bar{\phi}$ is an exponential moving average of the critic weights. The deterministic actor $\pi_\theta$ is learned by maximizing the expected returns

$$\mathcal{L}_\pi(\theta, \mathcal{D}) = \mathbb{E}_{\mathbf{o}_t \sim \mathcal{D}} \left[ Q_\phi(\mathbf{o}_t, \pi_\theta(\mathbf{o}_t)) \right]. \qquad (2)$$

## 4.2 Unsupervised RL Algorithms

As part of URLB, we open-source code for eight leading or well-known algorithms across all three of these categories all of which utilize the same optimization backbone. All algorithms provided with URLB differ only in their intrinsic reward while keeping all other parts of the RL architecture the same. We list all implemented baselines in Table 1 and provide a brief overview of the algorithms considered, which are binned into three categories – knowledge-based, data-based, and competence-based algorithms.[2] For detailed descriptions of each method we refer the reader to Appendix A.

*Knowledge-based Baselines:* Knowledge-based methods aim to increase knowledge about the world by maximizing prediction error. As part of the knowledge-based suite, we implement the Intrinsic Curiosity Module (ICM) [48], Disagreement [49], and Random Network Distillation (RND) [10]. All three methods utilize a function $g$ to either predict the dynamics $g(\mathbf{z}_{t+1}|\mathbf{z}_t, \mathbf{a}_t)$ (ICM, Disagreement) or predict the output of a random network $g(\mathbf{z}_t, \mathbf{a}_t)$ (RND), where $\mathbf{z}$ is the encoding of $\mathbf{o}$. ICM and RND maximize prediction error while Disagreement maximizes prediction uncertainty.

*Data-based Baselines:* Data-based methods aim to achieve data diversity by maximizing entropy. We implement APT [42] and ProtoRL [68] both of which maximize entropy $H(\mathbf{z})$ in different ways.

---

[2]We borrow this terminology from the following unsupervised RL tutorial [61].

Table 1: Unsupervised RL Algorithms implemented in URLB.

| Name | Algo. Type | Intrinsic Reward |
|---|---|---|
| ICM [48] | Knowledge | $\|g(\mathbf{z}_{t+1}\|\mathbf{z}_t, \mathbf{a}_t) - \mathbf{z}_{t+1}\|^2$ |
| Disagreement [49] | Knowledge | $\mathrm{Var}\{g_i(\mathbf{z}_{t+1}\|\mathbf{z}_t, \mathbf{a}_t)\} \quad i = 1, \ldots, N$ |
| RND [10] | Knowledge | $\|g(\mathbf{z}_t, \mathbf{a}_t) - \tilde{g}(\mathbf{z}_t, \mathbf{a}_t)\|_2^2$ |
| APT [42] | Data | $\sum_{j \in \mathrm{random}} \log \|\mathbf{z}_t - \mathbf{z}_j\| \quad j = 1, \ldots, K$ |
| ProtoRL [68] | Data | $\sum_{j \in \mathrm{prototypes}} \log \|\mathbf{z}_t - \mathbf{z}_j\| \quad j = 1, \ldots, K$ |
| SMM [40] | Competence | $\log p^*(\mathbf{z}) - \log q_\mathbf{w}(\mathbf{z}) - \log p(\mathbf{w}) + \log d(\mathbf{w}\|\mathbf{z})$ |
| DIAYN [21] | Competence | $\log q(\mathbf{w}\|\mathbf{z}) + \mathrm{const.}$ |
| APS [43] | Competence | $r_t^{\mathrm{APT}}(\mathbf{z}) + \log q(\mathbf{z}\|\mathbf{w})$ |

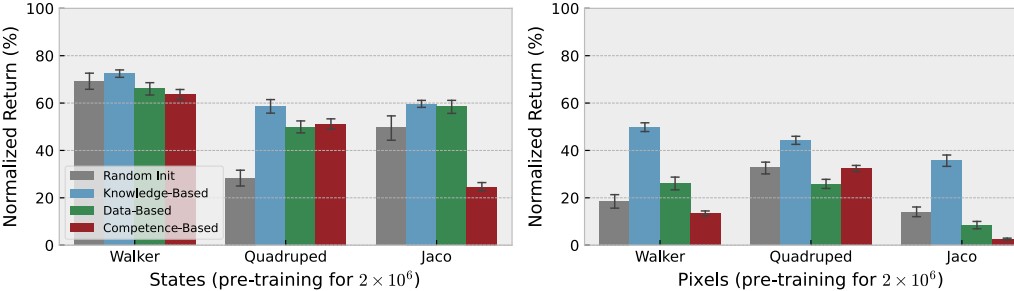

Figure 3: Aggregate results for each algorithm category after pre-training the agent with intrinsic rewards for 2M environment steps and finetuning with extrinisc rewards for 100k steps as described in Sec. 3.2. Scores are normalized by the asymptotic performance on each task (i.e., DrQ-v2 and DDPG performance after training from 2M steps on pixels and states correspondingly) and we show the mean and standard error of each category. Each algorithm is evaluated across ten random seeds. To provide an aggregate view of each algorithm category, the scores are averaged over individual tasks and methods (see Appendix C for detailed results for each algorithm and downstream task). The *Random Init* baseline represents DrQ-v2 and DDPG trained from a random initialization for 100k steps. Full results can be found in Section C.

Both methods utilize a particle estimator [60] to maximize the entropy by maximizing the distance between k-nearest neighbors (kNN) for each state or observation embedding $\mathbf{z}$. Since computing kNN over the entire replay buffer is expensive, APT estimates entropy across transitions in a randomly sampled minibatch. ProtoRL improves on APT by clustering the replay buffer with a contrastive deep clustering algorithm SWaV [12]. The centroids of the clusters are called *prototypes*, which are used by ProtoRL to estimate entropy.

*Competence-based Baselines:* Competence-based algorithms, learn an explicit skill vector $\mathbf{w}$ by maximizing the mutual information between the encoded observation and skill $I(\mathbf{z}; \mathbf{w})$. This mutual information can be decomposed in two ways, $I(\mathbf{z}; \mathbf{w}) = H(\mathbf{z}) - H(\mathbf{z}|\mathbf{w}) = H(\mathbf{w}) - H(\mathbf{w}|\mathbf{z})$. We provide baselines for both decompositions. The former decomposition is utilized in skill discovery algorithms such as DIAYN [21], VIC [24], VALOR [1], which are conceptually similar. For URLB, we implement DIAYN. The latter decomposition, though less common, is implemented in the APS [43], which uses a particle estimator for the entropy term and successor features to represent the conditional entropy [30]. Lastly, we implement SMM [40] which combines both decompositions into one objective. Note that the SMM paper describes both skill-based and skill-free variants, so it can be categorized as both competence and data-based.

## 5 Experiments

We evaluate the algorithms listed in Table 1 by pre-training with the intrinsic reward objective and fine-tuning on the downstream task as described in Section 3.2. For DrQ-v2 optimization we fix the hyper-parameters from [67] and for algorithm-specific hyper-parameters we perform a grid sweep

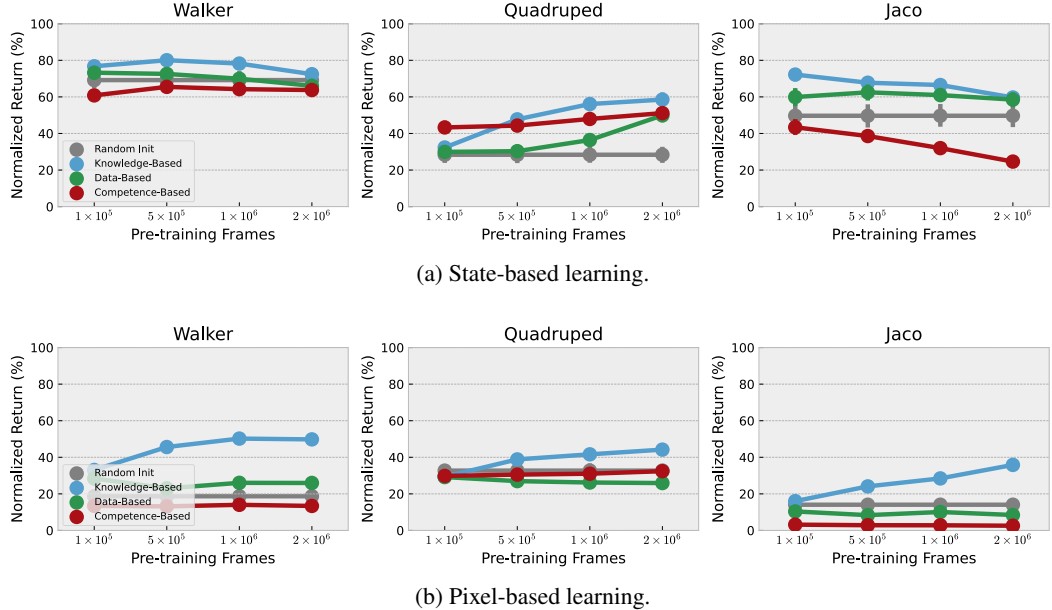

(a) State-based learning.

(b) Pixel-based learning.

Figure 4: We display the fine-tuning efficiency as a function of pre-training steps. As in Fig. 3 scores are asymptotically normalized, averaged across tasks and algorithms on a per-category basis, and evaluated over ten seeds. Our expectation is that a longer pre-training phase should lead to more efficient fine-tuning. However, in several cases the empirical evidence goes against our intuition demonstrating that longer pre-training is not always beneficial. Understanding this shortcoming of current methods is an important direction for future research. Detailed results can be found in Figures 6 and 7.

and pick the best performing parameters. We benchmark both state and pixel-based experiments and keep all non-algorithm-specific architectural details the same with a full description available in Appendix B. Performance on each downstream task is evaluated over ten random seeds and we display the mean scores and standard errors. We summarize the main results of our evaluation in Figures 3 and 4, which show evaluation scores grouped by algorithm category, described in Section 4.2, and environment, described in Section 3.3. An extensive list of results across all algorithms considered in this work can be found in Appendix C.

By benchmarking a wide array of exploration algorithms on both state and pixel-based tasks we are able to get perspective on the current state of unsupervised RL. Overall, we find that while unsupervised RL shows promise, it is still far from solving the proposed benchmark and many open questions need to be addressed to make progress toward unsupervised pre-training for RL. We note that solving the benchmark means matching the asymptotic DrQ-v2 (for pixels) and DDPG (for states) performance within 100k steps of fine-tuning. The motivation for this definition is that unsupervised RL agents get access to unlimited reward-free environment interactions. After pre-training, we seek to develop agents that adapt quickly to the desired downstream task. We list our observations below:

*O1: None of the implemented unsupervised RL algorithms solve the benchmark.* Despite access to up to 2M pre-training steps, after 100k steps of fine-tuning no method matches asymptotic performance on most tasks. The best-performing benchmarked algorithms achieve $40 - 70\%$ normalized return whereas the benchmark is considered solved when the agent achieves near $100\%$ normalized returns. This suggests that we are still far as a community from efficient generalization in deep RL.

*O2: Unsupervised RL is not universally better than random initialization.* We also observe that fine-tuning an unsupervised RL baseline is not always preferable to fine-tuning from a random initialization. In particular when learning from states, a random initialization is competitive with most baselines. However, when learning from pixels fine-tuning from random initialization degrades suggesting that representation learning is an important component of unsupervised pre-training.

*O3: There exists a large gap in performance between exploring from states and exploring from pixels.* Another observation that supports representation learning as an important aspect of exploration is that exploration algorithms degrade substantially when learning from pixels compared to learning from state. Shown in Figure 3, most algorithms lose $20 - 50\%$ when learning from pixels compared to state and especially so on the harder environments (Quadruped, Jaco Arm). These results suggest that better representation learning during pre-training is an important research direction.

*O4: In aggregate, competence-based approaches underperform knowledge-based and data-based approaches.* While knowledge-based and data-based approaches both perform competitively across URLB, we find that competence-based approaches are lagging behind. Specifically, there is no competence-based approach that achieves state-of-the-art mean performance on any of the URLB tasks, which points to competence-based unsupervised RL as an impactful research direction with significant room for improvement.

*O5: There is not a single leading unsupervised RL algorithm for both states and pixels.* We observe that there is no single state-of-the-art algorithm for unsupervised RL. At 2M pre-training steps, APT [42] and ProtoRL [68] are the leading algorithms for state-based URLB while ICM [48] achieves leading performance on pixel-based URLB despite the existence of more sophisticated knowledge-based methods [49, 10] (see Figure 5).

*O6: For many unsupervised RL algorithms, rather than monotonically improving performance decays as a function of pre-training steps.* We desire and would expect that the fine-tuning efficiency of unsupervised RL algorithms would improve as a function of pre-training steps. Surprisingly, we find that for 9 out of 18 experiments shown in Figure 4, performance either does not improve or even degrades as a function of pre-training steps. We see this as potentially the biggest drawback of current unsupervised RL approaches – they do not scale with the number of environment interactions. Developing algorithms that improve monotonically as a function of pre-training steps is an open and impactful line of research.

*O7: New fine-tuning strategies will likely be needed for fast adaptation.* While not investigated in depth in this benchmark, new fine-tuning strategies could play a large role in the adoption of unsupervised RL. Perhaps part of the issue raised in O6 could be addressed with better fine-tuning. The algorithms in URLB are all fine-tuned by initializing the actor-critic with the pre-trained weights and fine-tuning with an extrinsic reward. There are likely other better strategies for fine-tuning, particularly for competence based approaches that are conditioned on the skill $\mathbf{w}$.

# 6 Related work

**Deep Reinforcement Learning Benchmarks**. Part of the accelerated progress in deep RL over the last few years has been due to the existence of stable benchmarks. Specifically, the Atari Arcade Learning Environment [5], the OpenAI gym [7], and more recently the DeepMind Control (DMC) Suite [64] have become standard benchmarks for evaluating supervised RL agents in both state and pixel-based observation spaces and discrete and continuous action spaces. Open-sourcing code for algorithms has been another aspect that accelerated progress in deep RL. For instance, Duan et al. [19] not only presented a benchmark for continuous control but also provided baselines for common supervised RL algorithms, which led to the development of the widely used OpenAI gym benchmark [7] and baselines [18]. The combination of challenging yet feasible benchmarks and open-sourced code were important components in the discovery of many widely adopted RL algorithms [27, 44, 52–54].

In addition to Atari, OpenAi gym, and DeepMind control, there have been many other benchmarks designed to study different aspects of supervised RL. DeepMind lab [4] benchmarks 3D navigation from pixels, ProcGen [15, 16] measures generalization of supervised agents in procedurally generated environments, D4RL [22] and RL unplugged [26] benchmark performance of offline RL methods, B-Pref [39] benchmarks performance of preference-based RL methods, Metaworld [70] measures the performance of multi-task and meta-RL algorithms, and SafetyGym [51] measures how RL agents can achieve tasks with safety constraints. However, while the existing benchmarks are suitable for supervised RL algorithms, there is no such benchmark and collections of easy-to-use baseline algorithms for unsupervised RL, which is our primary motivation for accelerating progress in unsupervised RL through URLB.

**Unsupervised Reinforcement Learning**. While investigations into unsupervised deep RL appeared shortly after the landmark DQN [44], the field has experienced accelerated progress over the last year, which has been in part due to advents in unsupervised representation learning in CV [14, 31, 32] and NLP [8, 17, 50] as well as the development for stable RL optimization algorithms [27, 33, 41, 54]. However, unlike CV and NLP which focus solely on unsupervised representation learning, unsupervised RL has required both unsupervised representation and behavioral learning.

*Unsupervised Representation Learning for Deep RL:* In order for an RL algorithm to learn a policy $\pi(a|s)$ it must first have a good representation for the state $s$. When working with coordinate state, the representation is supplied by a the human task designer but when operating from image observations $o$, we must first transform the observations into latent vectors $z$. This transformation comprises the study of representation learning for RL. One of the first seminal works on unsupervised representation learning for RL showed that unsupervised auxiliary tasks improve performance of supervised RL [34]. Over the last two years, a series of works in unsupervised representation learning for RL with world models [28, 29] contrastive learning [38, 62, 68], autoencoders [66], and data augmentation [37, 67, 69] have dramaticaly improved learning efficiency from pixels. On many tasks from the DMC suite, learning from pixels is now as data-efficient as learning from state [38].

*Unsupervised Behavioral Learning for Deep RL:* One caveat is that the above algorithms are not fully unsupervised since they still optimize for an extrinsic reward but with an auxiliary unsupervised loss. Fully unsupervised RL also requires unsupervised learning of behaviors, which is typically achieved by optimizing for an intrinsic reward [47]. Given that representation learning is already heavily benchmarked for RL [28, 38, 69], URLB focuses mostly on unsupervised behavior learning. Many recent algorithms have been proposed for intrinsic behavioral learning, which include prediction methods [9, 48, 49], maximal entropy-based methods [11, 42, 43, 46, 56, 68], and maximal mutual information-based methods [21, 30, 43, 57]. However, these methods use different pre-training and evaluation procedures, different optimization algorithms, and different environments. To make fully unsupervised RL algorithm comparisons transparent and easier to develop, we introduce URLB.

## 7 Conclusion

We presented URLB, a benchmark designed to measure the performance of unsupervised RL algorithms. URLB consists of a suite of twelve evaluation tasks of varying difficulty from three domains and standardized procedures for pre-training and evaluation. We've open-sourced implementations and evaluation scores for eight leading unsupervised RL algorithms from all major algorithm categories. To minimize confounding factors, we utilized the same optimization method across all baselines. While none of the implemented baselines solve URLB, many make substantial progress suggesting a number of fruitful directions for unsupervised RL research. We hope that this benchmark makes the development and comparison of unsupervised RL algorithms easier and clearer.

**Limitations.** There are a number of limitations for both URLB and unsupervised RL methods in general. While URLB tasks are designed to be challenging, they are far from the visual and combinatorial complexity of real-world robotics. However, existing algorithms are unable to solve the benchmark meaning there is substantial room for improvement on the URLB tasks before moving on to even more challenging ones. While we present standardized pre-training and evaluation procedures, there can be many other ways of measuring the quality of the exploration algorithm. For instance, the quality of pre-training can be evaluated not only through policy adaptation but also through dataset diversity which we do not consider in this paper. In this work, similar to the Atari [5] and DMC [64] benchmarks for supervised RL we do not consider goal-conditioned RL which can be quite powerful for exploration [20]. For generality, we chose the currently most commonly used evaluation procedure that allowed us to benchmark a diverse set of leading exploration algorithms but, of course, other choices are available and would be interesting to investigate in future work.

**Potential negative impacts.** Unsupervised RL has the benefits of requiring zero extrinsic reward interactions during pre-training, and due to this the resulting agents may develop policies that are not aligned with human intent. This could be problematic in the long-term if not addressed early and carefully because as unsupervised robotics get more capable they can inadvertently inflict harm on themselves or the environment. Methods for constraining exploration within a broad set of human preferences (e.g. explore without harming the environment) is an interesting and important direction for future research in order to produced safe agents.

## Acknowledgements

This work was partially supported by Berkeley DeepDrive, BAIR, the Berkeley Center for Human-Compatible AI, the Office of Naval Research grant N00014-21-1-2769, and DARPA through the Machine Common Sense Program.

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
