# OpenReview forum: "URLB: Unsupervised Reinforcement Learning Benchmark"
_NeurIPS.cc/2021/Track/Datasets_and_Benchmarks/Round2 — NeurIPS 2021 Datasets and Benchmarks Track (Round 2)_

### Official Review · Reviewer_8jE6 · 2021-09-10
**An appropriate and interesting benchmark**

**Rating:** 7
**Confidence:** 5

**Strengths:**

This article provides a significant contribution towards simplifying comparative analysis between intrinsic motivation mechanisms, which is a very important step towards the development of robust multi-purpose learning architectures.

Code will be release as an open-source repository, which is good, and it appears to be properly documented.


**Weaknesses:**

## RIDE (potential major issue)

My main concern about this work is that the RIDE algorithm ("RIDE: Rewarding Impact-Driven Exploration for Procedurally-Generated Environments" Raileanu et. al, ICLR 2020), is not included in the benchmark. This algorithm was shown to outperform both RND and ICM, which are featured in URLB. I am willing to increase my score if authors manage to include RIDE into URLB during the rolling discussion, or if they provide a clear justification on why it does not make sense to include it.

## scripts for figures (minor)

Does your open source codebase contains scripts to reproduce easily your performance graphics ? Although minor, this is a very helpful addition for future benchmark users.

## aggregated graphics (minor comment)

A minor weakness is also the current status of the main body of the paper, which only reports aggregated results. Benchmark papers are often used by researchers for "shopping", i.e. to select the most efficient algorithm for their problem. I understand that given the breadth of the experimental campaign, aggregated graphics are needed. It is good that per-algorithm results are in the appendix. Besides, algorithm recommandations are made in observation 5 of the experimental section. That being said, I wonder whether adding something more visible in the conclusion might be helpful. Again, this is minor, feel free to keep the manuscript as is.

**Additional Feedback:**

About observation 3 of the experiments: you say that the performance gap from state-based vs pixel-based tasks is due to the quality of the exploration mechanism. Couldn't it just be a matter of the underlying DDPG-like agent which is less efficient at learning in pixel settings w.r.t. state settings ?

**Clarity:**

This paper is very well written.

I only noticed two typos:

* typo in figure 3: "extrinisc rewards" --> extrinsic
* l.284 "by a the human task designer" --> remove "a" or "the"

**Correctness:**

Evaluation methods and experiment design are both appropriate and well introduced

**Documentation:**

The repository appears to be properly documented and easy to use

**Relation To Prior Work:**

Related work is properly discussed.

Including "Large-Scale Study of Curiosity-Driven Learning" By Burda et al. might be appropriate.

**Summary And Contributions:**

Authors present URLB, a benchmark for unsupervised RL, enabling to compare algorithms producing intrinsic rewards.

The test procedure consists in 1) a pre-training phase on a given environment, without extrinsic rewards, then 2) a short fine-tuning phase for a specific task in the same environment (3 environments and 12 tasks are proposed based on DMlab), and finally 3) a post training performance assessment on this task.

Authors add 8 leading intrinsic motivation algorithms in the benchmark, and perform experiments with all of them, showing that URLB is an appropriate testbed for future work.

---

> ### Author Response · Authors · 2021-09-24
> **Thank you for your feedback. We address both questions raised by the reviewer - the RIDE baseline and scripts for reproducing results from the paper.**
>
> We thank the reviewer for the time spent reviewing our manuscript and the constructive feedback. We are glad the the reviewer found that “this article provides a significant contribution towards simplifying comparative analysis between intrinsic motivation mechanisms, which is a very important step towards the development of robust multi-purpose learning architectures.” Will be sure to fix all typos pointed out by the reviewer, and address their main concerns below:
>
>
> **Q1:** *My main concern about this work is that the RIDE algorithm is not included in the benchmark*
>
> RIDE is an exploration algorithm explicitly designed for procedurally generated environments, which tests a different kind of generalization than the one considered in this benchmark. Generalization in procedurally generated settings tests generalization to previously unseen observations, whereas URLB tests an agent’s ability to generalize to new behaviors.
>
> Additionally, it appears that the RIDE experiments are evaluated on Minigrid and the count-based bonus is counting unique raw frames. This works in Minigrid, because the observation space is small but for continuous states or high-dimensional images (e.g. DeepMind control or Atari) raw state counting is known not to work since all states are seen exactly once. RIDE does suggest to use pseudocounts in this instance, but their public code implementation only supports raw state counting (see this line of their code https://github.com/facebookresearch/impact-driven-exploration/blob/877c4ea530cc0ca3902211dba4e922bf8c3ce276/src/utils.py#L192).
>
> Because RIDE is built for procedurally generated environments and their pseudocount objective cannot be reproduced from their code release, it does not make sense to include it in URLB. However, if anyone would like to add RIDE (or any another unsupervised RL algorithm), they will be able to submit a Pull Request to URLB with their implementation and we will happily add it to the repository.
>
> **Q2:** *Does your open source codebase contains scripts to reproduce easily your performance graphics ? Although minor, this is a very helpful addition for future benchmark users.*
>
> Yes, each agent has a config file that specifies the hyperparameters we used. Simply running the default pretrain and finetune commands will reproduce all the results of the paper, since it will run the default hyperparameters specified by the config file.
>
> **Q3:** *Regarding pixel vs stat-based efficiency - couldn't it just be a matter of the underlying DDPG-like agent which is less efficient at learning in pixel settings w.r.t. state settings ?*
>
> Recent work on supervised RL (CURL, DrQ, RAD, DrQ-v2, ATC) has shown that RL from pixels can be as data-efficient as RL from state on many DeepMind control environments. Indeed, it could be that learning from pixels is just less efficient than from state, but given that supervised RL is as efficient (or close to being as efficient) as RL from state on DeepMind control, we would expect the gap to be smaller between state-based and pixel-based RL in URLB. This suggests that representation learning as a promising direction for research for unsupervised RL, and in particular understanding why representation learning schemes that are effective for supervised RL (e.g. data augmentations) are not as effective for unsupervised RL since the backbone DDPG algorithm is DrQ-v2 (a SOTA model-free algorithm for DeepMind control that is just DDPG + data augs when learning from pixels).
>
> **CURL** Laskin,  Michael,  Srinivas,  Aravind,  and  Abbeel,  Pieter.   Curl:  Contrastive  unsupervised representations for reinforcement learning. InInternational Conference on Machine Learning,2020.
>
> **RAD**  Laskin, Michael, Lee, Kimin, Stooke, Adam, Pinto, Lerrel, Abbeel, Pieter, and Srinivas, Aravind.Reinforcement learning with augmented data. InAdvances in Neural Information ProcessingSystems, 2020
>
> **DrQ** Yarats, Denis, Kostrikov, Ilya, and Fergus, Rob.  Image augmentation is all you need: Regu-larizing deep reinforcement learning from pixels.  InInternational Conference on LearningRepresentations, 2021
>
> **ATC** Stooke, Adam, Lee, Kimin, Abbeel, Pieter, and Laskin, Michael. Decoupling representation learning from reinforcement learning. InInternational Conference on Machine Learning, 2021
>
> **DrQ-v2** Yarats, Denis, Fergus, Rob, Lazaric, Alessandro, and Pinto, Lerrel. Mastering visual continuous control: Improved data-augmented reinforcement learning, 2021.

---

> > ### Comment · Reviewer_8jE6 · 2021-09-27
> > **Thank you for answering**
> >
> > Authors clarified my concern about not including RIDE: it seems logical not to include it, since vanilla RIDE do not handle pseudo-counts.
> >
> > I will raise my score to 7.
> >
> > Thank you for answering to my questions.

---

> > > ### Author Response · Authors · 2021-09-30
> > > **Thank you**
> > >
> > > Thank you for your feedback! We will be sure to incorporate the above clarifications in the manuscript and codebase readme.

---

### Official Review · Reviewer_b37q · 2021-09-17
**Relevant Benchmark for increasingly popular domain of unsupervised RL with some restrictions**

**Rating:** 7
**Confidence:** 5

**Strengths:**

The field of unsupervised RL has become increasingly popular with a wide array of evaluation protocols. Therefore, a unified evaluation protocol and benchmark appears well justified and valuable to the research community. The protocol appears sensible (even though potentially limiting -- see comments below) and open-source URL algorithms are provided which will help researchers to start working on these problems.

Also, the list of observations are appreciated as they can guide future research directions.

**Weaknesses:**

**Benchmark Protocol**

While fully unsupervised pre-training is certainly one interesting approach to leverage innovation in unsupervised RL, some approaches (in particular Knowledge-based baselines) are introduced to train with extrinsic and intrinsic rewards alongside each other. It might be worth considering such an evaluation setup as well (which seems like little additional implementation would be required).



**Evaluation:**

- Figure 4 lacks a point of reference for performance. I'd suggest to include a horizontal line indicating the normalised returns achieved by DDPG (with random initialisation/ without any pre-training) trained for the equivalent 100K steps in the testing task.
- Suggest to add standard deviation for normalised returns at each pre-training setting across all 4 tasks within the respective domain to indicate expected deviation across these runs.
- A precise explanation for the applied normalization scheme would be appreciated. It appears that asymptotic DDPG performance is the point of reference (l. 209) at 100% but this is not explicitely stated. Also, it might be valuable to give an indication how many steps of training are required for DDPG to achieve such asymptotic performance (presumably >> 100K) to get an understanding of the benefits achieved by unsupervised pre-training.



**Evaluation Observations**

- While these observations are certainly helpful, some lack substance. See the following examples
    - O4: do you have any insight into why competence-based approaches underperform? Are these approaches potentially not well suited for the evaluation protocol of unsupervised pre-training and then fine-tuning in separate tasks? Potentially they focus more on learning generalisable skills in contrast to enabling quick fine-tuning?
    - O6: Do you have any insight into why performance does not keep improving as more pre-training is executed? Are intrinsic rewards potentially degrading and thereby leading to a loss of previously learned skills or representations? It might be worth investigating the trend of intrinsic rewards achieved throughout training to investigate such questions.

**Additional Feedback:**

No additional comments.

**Clarity:**

The paper is written clearly.

One typo I noticed in l. 78f: "In this work, we pre-train agents with **intrinsic rewards $r^{ext}_t$ ** and fine-tune [...]" --> should be $r^{int}_t$

**Correctness:**

The claims made in the paper appear correct and sensible. For any criticism in the design of the benchmark and evaluation, see weaknesses section.



**Documentation:**

Basic documentation with instructions to run evaluations within the benchmark are provided. In-code documentation is minimal but likely sufficient.

Licensing is not stated within the paper, but the supplementary material indicates the MIT license is used.

It would be appreciated if the authors could briefly comment on intended maintenance for the benchmark. Are there any plans to further work on or extend the benchmark?

**Ethics:**

I do not foresee any ethical concerns that require further discussion.



**Relation To Prior Work:**

Related benchmarks and work within the domain of unsupervised RL are clearly stated. As mentioned in "Weaknesses", it might be worth expanding the benchmark to different settings applying unsupervised learning signals in RL outside of pre-training.



**Summary And Contributions:**

The paper introduces a new benchmark for unsupervised RL, evaluating the effectiveness of pre-training RL agents w varying self-supervised intrinsic rewards to enable improved sample efficiency in following fine-tuned tasks. The benchmark is building on top of the DeepMind Control Suite and contains a selection of twelve continuous control tasks as well as consistent implementations of eight unsupervised RL algorithms implemented on top of DDPG. An evaluation is presented for all tasks and provided algorithms with several observations to guide future research.

---

> ### Author Response · Authors · 2021-09-24
> **(Part 1 of author response) Thank you for your feedback. We address questions / clarifications raised by the reviewer.**
>
> We would like to thank you for the thorough review and feedback on how to improve the manuscript. We’re glad that the reviewer found that a “unified evaluation protocol and benchmark appears well justified and valuable to the research community.”
>
> We address your points of feedback below:
>
>
> **Q1:** *Some approaches (in particular Knowledge-based baselines) are introduced to train with extrinsic and intrinsic rewards alongside each other. It might be worth considering such an evaluation setup as well.*
>
> We agree that auxiliary unsupervised tasks can certainly help supervised RL agents learn more efficiently. However, the aim of this benchmark is to accelerate the development of RL algorithms that can generalize. To this end, fitting to any specific reward function during pre-training may hinder the agent’s generalization abilities to downstream tasks. One can of course pre-train with many reward functions, but that would substantially increase the design complexity of the benchmark (which rewards should we use for pre-training? How many? Sparse or dense? How should we balance exploration and exploitation (coefficient scheduling)?) and introduce a confounding factor independent of the intrinsic rewards the agents are currently trained with.
>
> This motivation is similar to that expressed in popular knowledge-based exploration papers. For instance, in Large Scale Curiosity (Burda et al.) the abstract states “However, annotating each environment with hand-designed, dense rewards is not scalable, motivating the need for developing reward functions that are intrinsic to the agent.” The aim of URLB is to test the quality of intrinsic reward functions disentangled from other factors that might affect performance.
>
> **Q2:** *Figure 4 lacks a point of reference for performance. I'd suggest to include a horizontal line indicating the normalised returns achieved by DDPG (with random initialisation/ without any pre-training) trained for the equivalent 100K steps in the testing task.*
>
> We agree and will add a horizontal bar on Figure 4 to provide a reference for performance. Thank you for the suggestion. See the last section of the content on the following link:
>
> https://anonymous.4open.science/r/neurips_urlb_rebuttal
>
>
> **Q3:** *Suggest to add standard deviation for normalised returns at each pre-training setting across all 4 tasks within the respective domain to indicate expected deviation across these runs.*
>
> We have provided such tables in the Appendix. Tables 4 & 5 in Appendix D provide the raw mean scores / standard errors for each algorithm, on each downstream task, for each pre-training snapshot, for both states and pixels. We’re happy to also add an expert-normalized score column to those tables to make them easier to compare to the aggregate results.

---

> > ### Author Response · Authors · 2021-09-24
> > **(Part 2 of author response) insight on competence methods + references**
> >
> >
> > **Q4:** *do you have any insight into why competence-based approaches underperform?*
> >
> > Yes, we found that competence-based methods for continuous control have been previously evaluated on OpenAI Gym tasks (see DIAYN). However, OpenAI gym continuous control tasks considered in prior work leak extrinsic signal into the environment by terminating the episode early if the agent loses balance. This means that the agent will naturally learn to stand, walk even without an exploration algorithm to prevent the episode from terminating early.
> >
> > To learn skills competence-based algorithms use a discriminator of the form $q(z|s)$. Since $s$ is constrained to meaningful behaviors in OpenAI Gym, $z$ will always map to a useful skill, However, in DeepMind control the episode length is fixed which means that no extrinsic signal leaks into the environment and an unsupervised exploration algorithm will discover many behaviors that are not useful for a downstream task (e.g. lying on the ground in different configurations). Now the skill vector $z$ is most likely to map to a degenerate set of uninteresting skills.
> >
> > Since the competence-based objective takes the form:
> >
> > $$I(s;z) = H(z) - H(z|s) \geq H(z) + \mathbb E [\log q(z|s) ]$$
> >
> > As long as $z$ is sampled randomly and $s$ maps to distinct $z$, the objective will be maximized even if the behaviors are not very structured or useful for downstream tasks. Thank you for the suggestion - we will add this discussion to the Appendix.
> >
> > **Q5:** *Do you have any insight into why performance does not keep improving as more pre-training is executed?*
> >
> > We found two reasons for this.
> >
> > 1. Intrinsic reward saturation - indeed some algorithms saturate their intrinsic reward (e.g. for APT on walker, the entropy starts to decrease). However, if the intrinsic reward is saturated we would expect the agent to adapt efficiently to the downstream task since it would have learned all possible behaviors. Instead, we observe that no methods are able to adapt to the downstream task efficiently even at saturation, which we hypothesize suggests that the environment has not been fully explored.
> >
> > 2. Finetuning strategies - an open research question (also raised in APT and APS papers) is how to finetune RL algorithms to downstream tasks. Currently, we finetune the DDPG actor-critic. However, the intrinsic and extrinsic reward functions are different and a critic that was trained to minimize the Bellman error on an intrinsic reward may not generalize when you swap in a new reward function. Given unsupervised RL is an emerging field, there is currently no agreed upon method for finetuning RL algorithms other than finetuning the weights. This is a very important and promising direction for future research.
> >
> > References:
> >
> > **Large-Scale Curiosity** Burda,  Yuri,  Edwards,  Harri,  Pathak,  Deepak,  Storkey,  Amos,  Darrell,  Trevor,  and Efros,Alexei A.   Large-scale study of curiosity-driven learning.   InInternational Conference onLearning Representations, 2019.
> >
> > **DIAYN** Eysenbach, Benjamin, Gupta, Abhishek, Ibarz, Julian, and Levine, Sergey.  Diversity is ally ou need: Learning skills without a reward function. InInternational Conference on LearningRepresentations, 2019.
> >
> > **APS** Liu, Hao and Abbeel, Pieter. APS: active pretraining with successor features. InInternationalConference on Machine Learning, 2021
> >
> > **APT** Liu, Hao and Abbeel, Pieter. Behavior from the void: Unsupervised active pre-training.arXivpreprint arXiv:2103.04551, 2021.

---

### Official Review · Reviewer_KgkF · 2021-09-21
**URLB: Unsupervised Reinforcement Learning Benchmark**

**Rating:** 6
**Confidence:** 4
**Correctness:** Looks good to me.

**Strengths:**

+ Good problem statement - making the development and comparison of unsupervised RL algorithms easier.
+ Open source benchmark with eight different algorithms in the benchmark
+ Baselines for the tasks are provided

**Weaknesses:**

- A bit of discussion about how the benchmark may evolve and how will it be kept up to date as the field evolves will be useful.
- Evaluation section could have been a bit more detailed.


**Additional Feedback:**

I would have appreciated more details on how the agents are learning. The results focus pretty much on the summarized end result but that intermittent training process is quite helpful to understand for readers/reviewers. I would have appreciated that as a reader.

One related topic/work that might be worth discussing is the implications of sim2real gap issues that generally plague RL. This doesn't come up anywhere, and it would be useful to discuss that.

In the presentation of the benchmark, I would have appreciated calling out the metrics a bit more in detail. Drawing out the attention that would be useful and important because a benchmark is more than just its tasks and algorithms. It is how we measure/assess "performance."


**Clarity:**

The paper is well written and the appendix provides all the necessary details about the experiments and evaluation.

**Documentation:**

The GitHub repo is good with basic documentation, however, I think more illustrative descriptions with some examples and code would be quite helpful for readers. This is just a minor suggestion.

**Ethics:**

Yes, this is discussed in the conclusion section where the focus is put on producing safe agents.

**Relation To Prior Work:**

All the relevant related work is fairly well described by the paper.

**Summary And Contributions:**

The paper focuses on evaluating unsupervised RL, specifically in the context where pre-training has been applied to agents so that it can help with generalization. Evaluating these methods has been challenging due to the lack of standard benchmarks and metrics. To this end, this paper introduces a benchmark for unsupervised RL that DeepMind's Control Suite. This benchmark is evaluated with existing algorithms and shows that we need more work in this space, which is a good motivation to use the benchmark to help solve this problem. The codebase is open source and publicly accessible.

---

> ### Author Response · Authors · 2021-09-24
> **Thank you for your feedback. We address all comments / concerns raised by the reviewer.**
>
> Thank you for your time spent reviewing our manuscript and the constructive feedback. We’re glad that you found our problem statement important, well motivated, and the codebase accessible. We address your points of feedback below:
>
> **Q1:** *A bit of discussion about how the benchmark may evolve and how will it be kept up to date as the field evolves will be useful.*
>
> We will continue actively supporting this benchmark over the coming months / years with a focus on two components:
>
> 1. Additional environments: We are planning on also supporting Atari and robotic manipulation environments in the future. For Atari, the main challenge is supporting multiple downstream tasks within each game. Almost all games in Atari support only one task. To change this, we would need to decode the RAM state per game and build custom tasks. This is possible but challenging and for this reason we opted for DeepMind Control as the first environment suite for this benchmark.
>
> 2. Additional algorithms: We will accept additional algorithms to be added to the benchmark which can be added by issuing a Pull Request with the algorithm, instructions for running, and results. This will enable anyone to add an algorithm to URLB and hopefully increase the quality and speed with which unsupervised RL algorithms are built.
>
> **Q2:** *​​Evaluation section could have been a bit more detailed.*
>
> Thank you for this point of feedback. We will be sure to add more detail to the evaluation section. Is there a specific aspect that you would like us to clarify?
>
> **Q3:** *I would have appreciated more details on how the agents are learning. The results focus pretty much on the summarized end result but that intermittent training process is quite helpful to understand for readers/reviewers.*
>
> We appreciate this point of feedback. Indeed, it’s insightful to provide intermittent training results. To do so, we’ve added learning curves for the finetuning phase which you can see in the link below:
>
> https://anonymous.4open.science/r/neurips_urlb_rebuttal
>
> **Q4:** *One related topic/work that might be worth discussing is the implications of sim2real gap issues that generally plague RL. This doesn't come up anywhere, and it would be useful to discuss that.*
>
> Thank you for this suggestion. However, the Sim2Real gap refers to a different type of generalization than the one that is the focus of this work. In this work, we want to train agents that adapt efficiently to downstream tasks within the same domain after unsupervised pre-training. In Sim2Real, we wish to learn representations that allow efficient transfer between simulated and real domains. The Sim2Real problem is interesting but is orthogonal to the generalization considered in this work and would likely require a different benchmark entirely. If this does not address your question sufficiently, could you please clarify how Sim2Real falls within the scope of this benchmark

---

### Official Review · Reviewer_PrUH · 2021-09-22

**Rating:** 7
**Confidence:** 4
**Clarity:** The paper is clear, albeit some choic…

**Strengths:**

Overall, this paper is well-written and it is a valuable contribution to facilitate unsupervised RL experiments, namely those where a representation is first pre-trained and then used for a downstream task. A few choices should be clarified in the author response; given these clarifications, I would be willing to increase the score.



**Weaknesses:**

As a benchmark paper, completeness and thoroughness in experimental design and description is key. One big omission I see is in discussing how hyperparameters are chosen for algorithms fairly. The main body says per-algorithm hyperparameters are tuned, with details in Appendix B. However, I only see a list of final hyperparameters. Which were swept, and how? Your approach can provide a general strategy for any work that uses this benchmark.

Similarly, the presentation of some of the results contrasting algorithms needs further explanation. It is important to explain why 10 seeds have been used, and why this is sufficient to make clear claims. Are the differences statistically significant? There are additionally some strong claims from the results here, that could be better qualified. For example, some of the differences could be due to using a base optimization algorithm of DDPG and how hyperparameters are chosen. Is it possible that conclusions about learning from pixel versus state are different if hyperparameters are different? Is this a general statement about these unsupervised losses? In general, quite a lot was run for this work, with some relatively strong conclusions; it could actually be better to run less and be more thorough in testing each question. If you do in fact think you have strong evidence for each claim, then it would be useful to add more justification for why you believe this experiment provides that evidence.

In this work, only the problem of continuous control is considered as part of the benchmark and only a certain variant of DDPG is considered as the optimization algorithm. How critical to the code-base is it that (a) DDPG be used and (b) the environments be for continuous control? It is understandable that a benchmark be reasonably scoped, but it would be useful to discuss if this choice is central. For example, do the authors think that representation learning for continuous control is a more interesting direction to pursue? Or was it simply a scoping choice that needed to be made?

Finally, some of the choices for the number of steps in learning should be better justified. What was the intuition behind selecting these specific values for pre-training and fine-tuning steps? It is possible that some algorithms might perform quite different for different such choices. Once again, it is not feasible to check all combinations, but an explanation for these choices is important. For example, maybe you choose them based on a the performance of a baseline algorithm.


**Additional Feedback:**

Minor comments:
1. Why is the standard error not visualized in figure 4 of the paper? They are important to visualize because the validity of 1-7 observations depends on them.
2. In line 79, “with intrinsic rewards $r^{ext}_t$” should be replaced by “with intrinsic rewards $r^{int}_t$.
3. The paper says: ““This suggests that we are still far as a community from efficient generalization in deep RL.” It is a bit strong of a claim. Does it need to be said?
4.It would be interesting to see the learning curve of the unsupervised approaches, especially during the fine-tuning process. It would be great to include these in the paper, even if just in the appendix.

**Correctness:**

The benchmark seems to be well-designed and the evaluation is mostly reasonable. However, as described under weaknesses, the work is a bit imprecise about certain choices that need clarification.

**Documentation:**

Yes.

**Ethics:**

No.

**Relation To Prior Work:**

There does not appear to be a discussion about other benchmarks. It seems natural to discuss some multi-task RL benchmarks, like Procgen. I understand that here the work is also about providing standardized implementations of the algorithms themselves.

**Summary And Contributions:**

This paper aims to standardize the process of benchmarking unsupervised reinforcement learning (RL) algorithms and sheds light on the limitations of the existing algorithms. To do so, it first proposes 12 benchmark tasks (i.e., 3 domains with 4 downstream tasks per domain) built upon the DeepMind Control Suite. Then, it introduces a methodology for pre-training, fine-tuning, and evaluation of these algorithms on the proposed environments. As baselines, 8 popular unsupervised RL algorithms are selected from the literature. It unifies the training and evaluation procedure of these baselines by keeping all of their components, such as the optimization scheme, the same and only varies the intrinsic rewards across these well-known algorithms. At last, the baselines are trained and evaluated on this set of environments based on the proposed standard procedure, and insights about the current state of the unsupervised RL methods are presented.

Edit: I have read the author response, and they have adequately addressed my concerns.

---

> ### Author Response · Authors · 2021-09-24
> **(Part 1 of author response) Thank you for your feedback. We address all questions and concerns raised by the reviewer.**
>
> We’d like to thank the reviewer for their time and feedback reviewing our manuscript. We’re glad that the reviewer found that this benchmark is a valuable contribution to the community to facilitate unsupervised RL experiments and provide answers for all questions raised below:
>
>  **Q1:** *Which hyperparameters were swept, and how? Your approach can provide a general strategy for any work that uses this benchmark.*
>
> One of our goals was to provide a fair comparison between the considered algorithms. To this end we were striving to unify the most important hyper-parameters (i.e. learning rate, batch size, etc.). These common hyper-parameters were selected using grid search. The algorithm-specific hyper-parameters were chosen from the original papers/implementations. However, several algorithms (for example ICM) required additional adaptation to accommodate continuous control, off-policy regime, and learning from pixel observations. In those cases we run additional grid searches to identify better hyper-parameters.
>
>
>
> **Q2:** *It is important to explain why 10 seeds have been used, and why this is sufficient to make clear claims. Are the differences statistically significant?*
>
> Thank you for the question. Benchmarking with 10 seeds on DeepMind Control is the standard evaluation protocol in deep RL over the last couple of years (SAC-AE, PytorchSac, CURL, DrQ, RAD, PlaNet, Dreamer). The rationale for why 10 seeds is statistically significant has been recently shown in RLiable, where the authors show empirically that results from stratified bootstrap estimates do not vary much after 10 seeds. To further support this point, we provide stratified bootstrap estimates (see RLiable) of aggregate statistics (mean, median, interquartile mean, optimality gap) using the protocol introduced in RLiable for reproducible RL. In the link below int the **Aggregate Statistics** section, you can see that the aggregate results look nearly identical to those reported in our manuscript.
>
> https://anonymous.4open.science/r/neurips_urlb_rebuttal
>
> **Q3:**  *Is it possible that conclusions about learning from pixel versus state are different if hyperparameters are different? Is this a general statement about these unsupervised losses? *
>
> The hyper-parameters for both settings were chosen to achieve the best possible performance for each algorithm. To the best of our current knowledge, improving the current performance results would require further algorithmic advancements, rather than optimizing hyper-parameters. That is why we believe this benchmark can be useful to advance our understanding about Unsupervised RL.
>
>
> **Q4:** *How critical to the code-base is it that (a) DDPG be used and (b) the environments be for continuous control? Or was it simply a scoping choice that needed to be made?*
>
> We chose DDPG as a learning algorithm based on recent results from DrQ-v2 (Yarats at al. 2021), which demonstrated that in image-based settings replacing the SAC backbone of DrQ with DDPG leads to better performance. Similarly, we observed that DDPG implementation from DrQ-v2 outperforms SAC in state-based settings. As our goal was to choose the best available algorithm for continuous control we picked DrQ-v2 (for pixels) and DDPG from DrQ-v2 (for states). We will update the paper to explain this rational.
>
> **[Continued in next comment]**

---

> > ### Author Response · Authors · 2021-09-24
> > **(Part 2 of author response) includes references**
> >
> >
> > **Q5:** *Some of the choices for the number of steps in learning should be better justified. What was the intuition behind selecting these specific values for pre-training and fine-tuning steps?*
> >
> > Thank you for pointing out that we could have justified some design choices more clearly. In unsupervised RL, it is common to have extended pre-training phases (e.g. up to 250m steps - see APS for example) followed by efficient finetuning phases (e.g. 100k steps). We chose to upper bound pre-training with 2M steps simply to make this benchmark accessible to a wide array of RL researchers. For context, training for 250M steps requires compute resources usually only available in industry lab settings. On DeepMind control, a supervised DDPG agent is able to solve most tasks in URLB by 2M steps, which was the heuristic choosing this upper bound.
> >
> > For finetuning, the 100k benchmark has already been established as the data-efficient benchmark for DeepMind control by a number of prior works (CURL, RAD, DrQ). There are two reasons for this. First, 100k corresponds to ~2 hours of real-time training, which is a rough estimate of the amount of time it would take a human to learn a new skill. The second reason is that at 100k steps, no supervised RL algorithms can solve the downstream task optimally so there is significant room for improvement. If we were to consider a less data-efficient benchmark (e.g. 500k, 1M, or 2M finetuning steps) the finetuned algorithms would start achieving near-optimal scores and the comparisons would not be meaningful since there is little room for improvement.
> >
> > We’ve provided learning curves for up to 200k finetuning steps below to show that the algorithms finetune monotonically, meaning that choosing a different number of finetuning steps would not impact the results significantly.
> >
> > https://anonymous.4open.science/r/neurips_urlb_rebuttal
> >
> >
> > **Q6:** *There does not appear to be a discussion about other benchmarks.*
> >
> > We refer the reviewer to the subsection titled “Deep Reinforcement Learning Benchmarks” in the Related Work section which discusses prior RL benchmarks and how URLB is different.
> >
> > References:
> >
> > **RLiable** Rishabh Agarwal, Max Schwarzer, Pablo Samuel Castro, Aaron Courville, and Marc G. Bellemare. Deep reinforcement learning at the edge of the statistical precipice, 2021.
> >
> > **DrQ-v2** Yarats, Denis, Fergus, Rob, Lazaric, Alessandro, and Pinto, Lerrel. Mastering visual continuous control: Improved data-augmented reinforcement learning, 2021.
> >
> > **CURL** Laskin,  Michael,  Srinivas,  Aravind,  and  Abbeel,  Pieter.   Curl:  Contrastive  unsupervised representations for reinforcement learning. InInternational Conference on Machine Learning,2020.
> >
> > **RAD**  Laskin, Michael, Lee, Kimin, Stooke, Adam, Pinto, Lerrel, Abbeel, Pieter, and Srinivas, Aravind.Reinforcement learning with augmented data. InAdvances in Neural Information ProcessingSystems, 2020
> >
> > **DrQ** Yarats, Denis, Kostrikov, Ilya, and Fergus, Rob.  Image augmentation is all you need: Regu-larizing deep reinforcement learning from pixels.  InInternational Conference on LearningRepresentations, 2021

---

> > > ### Comment · Reviewer_PrUH · 2021-09-30
> > > **Thank you for answering all my questions quite comprehensively**
> > >
> > > Most of my concerns have been allayed, and I've increased my score to a 7.
> > >
> > > I hope you add some of these clarifications and justifications to the paper. They were very helpful! A benchmark papers makes important design decisions, and for follow-up work using the benchmark, having those design decisions explicit will make it more likely for the benchmark to be used appropriately and for appropriate conclusions to be drawn.
> > >
> > > Finally, I actually realized upon second reading that the term "unsupervised RL algorithms" was a bit confusing, when really you only change the intrinsic reward. I think this could be made clearer up front. Some people coming to this work might think the unsupervised part is about self-supervision in general (eg to learn representations). Clarifying this up front would be good, or even using terminology that is a bit more specific. I understand the community uses this term, so it would make little sense to deviate. But, it could be useful to say something like "unsupervised RL algorithms, where the agent learns using intrinsic rewards without extrinsic rewards" the first time the term is used.

---

> > > > ### Author Response · Authors · 2021-09-30
> > > > **Thank you**
> > > >
> > > > This was very helpful feedback, thank you! We will be sure to add these clarifications to the manuscript and appendix as well as clarifying the meaning of "unsupervised RL" up front.

---

### Official Review · Reviewer_ZrB4 · 2021-09-22
**Problems with how all algorithms are evaluated**

**Rating:** 6
**Confidence:** 3

**Strengths:**

I agree with the authors that establishing unified benchmark for unsueprvised RL is necessary. The paper is well-motivated.
The paper is well-written and easy to follow.
I like the part where the authors grouped algorithms and organized the evaluation into state-based and pixel-based scenarios.
The number of comparative methods is sufficient and covers different types of RL approaches.

**Weaknesses:**

I have concerns about the evaluation metrics and the way experiments are designed. I would be willing to modify the scores if the authors could explain or justify the following:

1. What is normalized return? normalize with respect to which algorithm? Why not directly show the absolute return over all runs? I looked into table 4 and 5 in the appendix. I am still confused about what the individual results mean. Are those "averaged returns over runs" or anything else?

2. In the paper, the authors evaluated all the algorithms using overall normalized return. What about other evaluation metrics for individual algorithms, e.g. plots of intrinsic rewards for knowledge-based methods or the mutual information plots for competence-based methods or predictive errors for generative approaches?

3. I am curious about why authors only evaluate fine-tuned models at 100k steps. Different methods have different numbers of parameters (due to architecture differences, replay buffer sizes, and so on). I am not sure whether it is a fair comparison to only report performance at fixed 100k steps. And why 100k?

4. When authors refer to unsupervised representation learning, I wonder which environments are all the models trained on for unsupervised representation learning. E.g. are they trained on the same environment where the fine-tuned environments are. This is related to testing the generalization ability of these algorithms. If not, how similar all these environments in terms of representation learning. Similarly, how similar are the behavioral learning tasks at unsupervised and fined-tuned stages?

5. I like the point that authors disentangle representation and behavioral learning. However, I am not sure why each unsupervised learning RL algorithm should perform consistently well in both behavioral or representation learning, as one of the observations in the Experiment Section verified this point. Why is it necessary to compare these for the same algorithm in the first place?

6.  I found that all the observations in Sec Experiments are interesting. It would be great if authors could provide explanations about why different types of algorithms behave so differently.

Minor:
1. line 79: notation for intrinsic reward is wrong: r^{int}_t
2. The generated PDF files for the main paper and supplementary material are only viewable online using browsers. The downloaded version (unzip) causes errors when opening PDF offline.

**Additional Feedback:**

Please see the weakness section.

**Clarity:**

The paper is well-written and well-structured. The motivation is well-explained.


**Correctness:**

The source code for the implementation of each algorithm is available. I believe that the evaluations are implemented correctly; though I did not run the code by myself.

I have several concerns about the design of the evaluation methods and the way all algorithms are compared (see weakness).

**Documentation:**

The source code for the implementation of each algorithm is available. All implementations including hyperparameters are provided in the main paper and supplementary material. I do not foresee any ethical issues. I agree with the authors about the negative impacts that might exist in real-world applications.

**Ethics:**

As far as I can see, no ethical issues

**Relation To Prior Work:**

There is sufficient coverage of related works.

**Summary And Contributions:**

The authors introduced the Unsupervised Reinforcement Learning Benchmark. 12 continuous control tasks from three domains are studied in DeepMind control suite. The eight baselines are compared. The source code is publicly available.

---

> ### Author Response · Authors · 2021-09-24
> **(Part 1 of author response) Thank you for your feedback. We address all of the questions and concerns raised by the reviewer.**
>
> Thank you for your time and constructive feedback reviewing URLB! We’re glad that you found the benchmark to be well motivated, the paper to be well written, and the experiments to be sufficient.
>
> We address each of the posed questions below and will be sure to revise the manuscript to incorporate the suggested clarifications:
>
> **Q1:** *What is normalized return?*
>
> Normalized return is the return divided by the return achieved by an expert policy (expert return). The expert return is defined as the score achieved by a DDPG agent after 2M steps of training with extrinsic rewards. For the tasks presented in this benchmark, a DDPG is able to learn the optimal policy for each task by 2M steps of training (20x more steps than the 100k finetuning budget).
>
> **Q2:** *Algorithms evaluated using overall normalized return. What about other evaluation metrics for individual algorithms?*
>
> Similar to unsupervised learning evaluations in Computer Vision and NLP, we evaluate unsupervised pre-training for RL with respect to how efficiently a pre-trained agent adapts to a downstream task. Ultimately we desire RL agents to generalize quickly to new tasks and this is the most direct way of measuring how quickly pre-trained agents can generalize. Other metrics like mutual information are also interesting but they are hard to compare. For instance, mutual information methods use different estimates for a variational lower bound, so direct comparisons between different lower bounds are not meaningful. Another example is that knowledge-based methods optimize different objectives - ICM maximizes prediction error of a dynamics model, Disagreement maximizes uncertainty, while RND maximizes prediction error of a random network encoding. Comparing using these intrinsic reward metrics is not meaningful since each algorithm uses a different metric.
>
> **Q3:** *I am curious about why authors only evaluate fine-tuned models at 100k steps.*
>
> We choose 100k steps for two reasons. First, it is already an accepted data-efficient benchmark for DeepMind Control (CURL, RAD, DrQ). The motivation for this is that 100k steps corresponds to ~2hrs of real-time training, which is how long we would expect a human to acquire a new skill (see SimPLe). Second, at 100k steps, there is a lot of room for improvement. Supervised RL algorithms do not learn the optimal policy for most tasks until 2M training steps. For this reason, if we can pre-train agents and adapt them to solve downstream tasks in 100k steps (which would solve URLB) that would improve the generalization efficiency of RL by 10-20x which would be an exciting advance. Moreover, for DeepMind Control the asymptotic performance is bounded to a score of 1000. If we were to consider a less data-efficient benchmark (e.g. 500k, 1M, or 2M finetuning steps) the finetuned algorithms would start achieving near-optimal scores and the comparisons would not be meaningful since there is little room for improvement.
>
> We’ve provided learning curves for up to 200k finetuning steps below to show that the algorithms finetune monotonically, meaning that choosing a different number of finetuning steps would not impact the results significantly.
>
> https://anonymous.4open.science/r/neurips_urlb_rebuttal
>
> **Q4:** *Different methods have different numbers of parameters (due to architecture differences, replay buffer sizes, and so on).*
>
> For all algorithms in URLB we re-implemented them to keep the architecture / parameter count fixed as much as possible across methods. All algorithms use a replay buffer of the same size, same encoder architecture, same batch size, and are generally identical except for the intrinsic reward used. The overall parameter count is roughly the same for each method, since auxiliary task networks (forward model, entropy prediction, mutual info estimation) use similar architectures.  For common hyper-parameters (learning rate, batch size, replay buffer size) we performed a grid search and picked the set of parameters that worked best across all methods. For algorithm-specific hyperparameters (e.g. dynamics model network) we either used the hyperparameters presented in the original papers and performed a grid sweep to pick the best set of hyperparameters for each algorithm. For these reasons, performance differences between the methods should not be due to architectural / hyperparameter differences but rather the quality of the intrinsic reward function.
>
> **Q5:** *When authors refer to unsupervised representation learning - are the algorithms trained on the same environment where the fine-tuned environments are?*
>
> Yes, each agent is first pre-trained on a domain (e.g. quadruped) and the finetuned on tasks within that domain (e.g. quadruped walk, run, etc). Each agent is pre-trained by maximizing its intrinsic reward, which has no direct similarity to the behavior of the downstream tasks.
>
> **[Continued in next comment]**

---

> > ### Author Response · Authors · 2021-09-24
> > **Part 2 of author response (includes references for Part 1)**
> >
> > **Q6:** *I am not sure why each unsupervised learning RL algorithm should perform consistently well in both behavioral or representation learning, as one of the observations in the Experiment Section verified this point.*
> >
> > The gap between state and pixel-based performance described in O3 of the Experiments section was surprising to us for two reasons. First, better behavioral exploration should lead to better representation learning due to increased data diversity. This effect was previously shown in ATC (Stooke et al) where representations learned from random exploration were much worse than those learned from a policy with more diverse behaviors. Second, prior work (DrQ, CURL, RAD) has shown that supervised RL from pixels can be as data-efficient as RL from state on DeepMind control through the use of representation learning and data augmentation. For these reasons, we would expect unsupervised RL from state and pixels to be comparable in data-efficiency on DeepMind control and were surprised to find the gap was so big suggesting representation learning as a promising direction for future research.
> >
> > **Q7:** *I found that all the observations in Sec Experiments are interesting. It would be great if authors could provide explanations about why different types of algorithms behave so differently.*
> >
> > Thank you for the suggestion. We’ll add a section to the appendix providing the discussion below:
> >
> > Across the three methods - data-based, knowledge-based, and competence-based - the best data-based and knowledge-based methods are competitive with one another. For instance, RND (a leading knowledge-based methods) and ProtoRL (a leading data-based method) achieve similar finetuning scores. Both are maximizing data diversity in two different ways - one through maximizing prediction error and the other through entropy maximization.
> >
> > On the other hand, competence-based methods as a whole do much worse than data-based and knowledge-based ones. We hypothesize that this is due to current competence-based methods only supporting small skill spaces. Competence-based methods maximize a variational lower bound to the mutual information of the form:
> >
> > $$ I(s;z) = H(z) - H(z|s) = H(z) + \mathbb E [ \log p(z|s) ] \geq H(z) + \mathbb [\log q(z|s)] $$
> >
> > where $q(z|s)$ is called the discriminator. The discriminator can be interpreted as a classifier from $s \rightarrow z$ (or vice versa depending on how you decompose $I(s;z)$ ). In order to have an accurate discriminator, $z$ is chosen to be small in practice (DIAYN - $z$ is a 16 dim one-hot, SMM - $z$ is 4 dim continuous, APS - $z$ is 10 dim continuous).
> >
> > OpenAI gym environments for continuous control mask this limitation because they terminate if the agent falls over and hence leak extrinsic signal about the downstream task into the environment. This means that the agent learns only useful behaviors that keep it balanced and therefore a small skill vector is sufficient for classifying these behaviors. However, in DeepMind control (and hence URLB) the episodes have fixed length and therefore the set of possible behaviors is much larger. If the skill space is too small, the most likely skills to be classified are different configurations of the agent lying on the ground. We hypothesize that building more powerful discriminators would improve competence-based exploration.
> >
> > References:
> >
> > **CURL** Laskin,  Michael,  Srinivas,  Aravind,  and  Abbeel,  Pieter.   Curl:  Contrastive  unsupervised representations for reinforcement learning. InInternational Conference on Machine Learning,2020.
> >
> > **RAD**  Laskin, Michael, Lee, Kimin, Stooke, Adam, Pinto, Lerrel, Abbeel, Pieter, and Srinivas, Aravind.Reinforcement learning with augmented data. InAdvances in Neural Information ProcessingSystems, 2020
> >
> > **DrQ** Yarats, Denis, Kostrikov, Ilya, and Fergus, Rob.  Image augmentation is all you need: Regu-larizing deep reinforcement learning from pixels.  InInternational Conference on LearningRepresentations, 2021
> >
> > **ATC** Stooke, Adam, Lee, Kimin, Abbeel, Pieter, and Laskin, Michael. Decoupling representation learning from reinforcement learning. InInternational Conference on Machine Learning, 2021
> >
> > **DIAYN** Eysenbach, Benjamin, Gupta, Abhishek, Ibarz, Julian, and Levine, Sergey.  Diversity is ally ou need: Learning skills without a reward function. InInternational Conference on LearningRepresentations, 2019.
> >
> > **SMM** Lee, Lisa, Eysenbach, Benjamin, Parisotto, Emilio, Xing, Eric P., Levine, Sergey, and Salakhut-dinov, Ruslan.   Efficient exploration via state marginal matching.CoRR, abs/1906.05274,2019
> >
> > **APS** Liu, Hao and Abbeel, Pieter. APS: active pretraining with successor features. InInternationalConference on Machine Learning, 2021

---

> > > ### Comment · Reviewer_ZrB4 · 2021-09-25
> > > **Thank you for the replies**
> > >
> > > The authors have adequately addressed my concerns. Given a clearer picture of the evaluation methods and insights, I would like to update my score from 5 to 6 on the premise that I trust that the authors would improve the current version of the manuscript by incorporating the comments and clarifications as they promised in the rebuttal.

---

> > > > ### Author Response · Authors · 2021-09-30
> > > > **Thank you**
> > > >
> > > > Thank you for your valuable feedback. We will be sure to incorporate these comments in our manuscript by adding clarifications in the main text as well as expanding the appendix.

---

### Decision · Program_Chairs · 2021-10-10

**Decision:**

Accept

**Comment:**

This paper proposes a benchmark for evaluating unsupervised RL agents - i.e. agents that learn from intrinsically generated reward signals or other kinds of pre-training before being exposed to the final task. Developing such endogenous RL agents is important, however progress is hard to measure without a comprehensive benchmark, so this submission is very well placed. All of the reviewers agreed that this is a solid submission, evaluating a large number of algorithms on a good set of tasks. For this reason I recommend acceptance.